# Synergistic Upregulation of Target Genes by TET1 and VP64 in the dCas9–SunTag Platform

**DOI:** 10.3390/ijms21051574

**Published:** 2020-02-25

**Authors:** Sumiyo Morita, Takuro Horii, Mika Kimura, Izuho Hatada

**Affiliations:** Laboratory of Genome Science, Biosignal Genome Resource Center, Institute for Molecular and Cellular Regulation, Gunma University, 3-39-15 Showa-machi, Maebashi 371-8512, Japan; msumiyo@gunma-u.ac.jp (S.M.); horii@gunma-u.ac.jp (T.H.); mikimura@gunma-u.ac.jp (M.K.)

**Keywords:** CRISPR/Cas9, dCas9, SunTag, Tet1, VP64

## Abstract

Overexpression of a gene of interest is a general approach used in both basic research and therapeutic applications. However, the conventional approach involving overexpression of exogenous genes has difficulty achieving complete genome coverage, and is also limited by the cloning capacity of viral vectors. Therefore, an alternative approach would be to drive the expression of an endogenous gene using an artificial transcriptional activator. Fusion proteins of dCas9 and a transcription activation domain, such as dCas9–VP64, are widely used for activation of endogenous genes. However, when using a single sgRNA, the activation range is low. Consequently, tiling of several sgRNAs is required for robust transcriptional activation. Here we describe the screening of factors that exhibit the best synergistic activation of gene expression with TET1 in the dCas9–SunTag format. All seven factors examined showed some synergy with TET1. Among them, VP64 gave the best results. Thus, simultaneous tethering of VP64 and TET1 to a target gene using an optimized dCas9–SunTag format synergistically activates gene expression using a single sgRNA.

## 1. Introduction

Overexpression of a gene of interest is a general approach used in both fundamental research and therapeutic applications. Conventionally, overexpression uses an exogenous gene. In basic research, exogenous expression of a genome-wide cDNA library is the most commonly used gain-of-function approach for the systematic elucidation of gene function. In a therapeutic context, exogenous expression of several key transcription factors in differentiated cells leads to a transition in cellular state and generates induced pluripotent stem cells (iPSCs) [1]. Exogenous expression of a functional copy by a viral vector, such as an adeno-associated virus (AAV) vector, can be used in gene therapy to replace mutant genes, thereby treating human diseases caused by haploinsufficiency [2]. However, this approach is inefficient and has two crucial limitations. First, it is difficult to construct a cDNA library that covers the whole genome, in particular in a manner that encompasses isoform variance. Second, large cDNA sequences are often difficult to clone into size-limited viral expression vectors.

One alternative approach for overexpression is activation of endogenous genes by an artificial transcriptional activator consisting of a programmable DNA-binding protein and an effector domain. In this method, the programmable DNA-binding protein tethers the transcription activation domains to the promoter of the target gene [3,4,5,6,7,8,9]. The following are three types of DNA-binding proteins currently in wide use: zinc finger proteins [10], transcription activator-like effectors (TALEs) [11], and catalytically dead Cas9 (dCas9) [12,13,14]. Among these, dCas9 is the most appropriate for genome-wide gain-of-function screening due to its simplicity and versatility.

Fusions of dCas9 and a transcription activation domain (e.g., dCas9–VP64) are widely used for the activation of endogenous genes. In this approach, expression is induced by targeting the promoter region of the gene of interest. Programmable targeting of endogenous loci can be achieved by an individual single-guide RNA (sgRNA) [7,8,9]. However, because the activation range is low when only a single sgRNA is used, tiling of the target promoter region with several sgRNAs is often necessary to achieve robust transcriptional activation [7,8,9]. Consequently, it is difficult to use this format for genome-wide gain-of-function screening and gene therapy. To solve this problem, a format that recruits multiple factors to dCas9, thereby improving the activation range, was recently developed [15]. The synergistic activation mediator (SAM) system consists of dCas9–VP64, modified gRNA containing RNA aptamers, and MS2 bacteriophage coat protein fused to a transcription activation domain p65–HSF1. VP64 fused to dCas9 and MS2–p65–HSF1 recruited to dCas9 by the RNA aptamers synergistically activates transcription of the target gene.

In previous work, we developed a modified dCas9–SunTag format for efficient, targeted demethylation and activation of specific DNA loci using a single sgRNA [16]. SunTag is a repeated GCN4 peptide tag array that recruits multiple copies of an effector molecule fused to anti-GCN4 peptide antibody (scFv) [17]. We optimized the linker length between the tags to maximize transcriptional activation by the effector TET1, which performs the first enzymatic steps during DNA demethylation.

Here, we have described simultaneous tethering of VP64 and TET1 to a target gene using the optimized dCas9–SunTag format, resulting in synergistic activation of gene expression.

## 2. Results

### 2.1. Selection of Another Factor that Activates Gene Expression Synergistically with TET1 in the Modified dCas9–SunTag Format

In the modified dCas9–SunTag format, dCas9 is fused to a tandem array of multiple copies of a 19-amino acid (aa) GCN4 peptide tag separated by a 22-amino acid linker [16]. This format can recruit multiple copies of anti-GCN4 peptide antibody (scFv)-fused catalytic domain of TET1 (scFv-TET1) to the target promoter, leading to demethylation and activation of the gene. If anti-GCN4 peptide antibody (scFv) that is fused to another factor (scFv–Factor X) is used along with scFv–TET1, then both TET1 and Factor X can be recruited to the target with a single sgRNA (Figure 1) and should synergistically activate the target gene. Therefore, we used this format to search for other factors that synergistically activate gene expression, when used in conjunction with TET1.

We investigated the synergy between TET1 and other factors, hereafter referred to as X. Candidate factors were transcription factors (VP64 and p65HSF1), a coactivator (p300), a putative SWI/SNF subunit (SS18), a heterochromatin relaxer (GADD45A), and pioneer factor factors (FOXA1 and PU.1). Ten genes (CARD9, KDM2B, RAB19, CNKSR1, SBNO2, SPARC, CLEC11A, HGF, TCF21, and TINAGL1) were examined for each factor. All these genes are hypermethylated in the A549 lung adenocarcinoma cell line. Expression levels were normalized against the corresponding level of actin mRNA (Appendix A) and were presented as fold changes relative to GFP-transfected experimental controls (Figure 2). When X was VP64, p65HSF1, p300, SS18, GADD45A, FOXA1, or PU.1, cells were transfected with dCas9–SunTag and scFv–TET1 (Figure 2, green bar); dCas9–SunTag and scFv–X (Figure 2, blue bar); or dCas9–SunTag, scFv–TET1 and scFv–X (Figure 2, red bar). Synergy between TET1 and X was judged as follows. The expression level in cells transfected with dCas9–SunTag and scFv–X (Figure 2, blue bar) were compared with the level in cells transfected with dCas9–SunTag and scFv–TET1 (Figure 2, green bar), and the cells that exhibited a higher expression level were selected. These expression levels were compared with those of cells transfected with dCas9–SunTag, scFv–TET1, and scFv–X (Figure 2, red bar). The results are summarized in Figure 3. The SunTag system with only TET1 yielded 1.5- to 78-fold upregulation (average: 20-fold; Figure 2, green bar). On the other hand, the SunTag system with TET1 and VP64 yielded 3.5- to 1139-fold upregulation (average: 212-fold; Figure 2, red bar). In eight of ten genes examined, synergy between TET1 and VP64 was observed at a rate of 1.8- to 21-fold (average: 6.5-fold; Figure 2 and Figure 3). Transcription factor p65HSF1 exhibited synergy with TET1 in five of ten genes examined. Other factors exhibited synergy with TET1 in no more than three genes. Thus, relative to other factors, transcription factors yielded a stronger synergistic effect.

### 2.2. Comparison of Synergistic Effects between Modified SunTag and dCas9 Direct Fusion

Direct fusions of dCas9 and a transcription activation domain, such as dCas9–VP64, are widely used for the activation of endogenous genes [7,8,9]. However, the activation range is low when an individual single-guide RNA (sgRNA) is used. Activation by dCas9–VP64 is improved when it is used with dCas9-TET1 [18]. We compared the activation level of this direct fusion system with that of our SunTag system. Using the same single sgRNA for each gene, we compared the expression levels in cells transfected with dCas9–SunTag, scFv–TET1, and scFv–VP64, with those in cells transfected with dCas9–TET1 and dCas9–VP64. For nine of ten genes examined, the SunTag system yielded significantly better activation (Figure 4).

## 3. Discussion

In the modified dCas9–SunTag format, multiple factors can be recruited to the target gene to synergistically activate transcription (Figure 1). We sought to identify the factor that exhibited the best synergy with TET1 in the dCas9–SunTag format. All seven of the factors that we tested showed some synergy with TET1 in at least one of ten genes examined. Among them, VP64 gave the best results, exhibiting a synergistic effect with TET1 in eight of ten genes examined (Figure 2 and Figure 3). Although it remains unclear why VP64 gave the best synergy, it may be due to the fact that VP64 had the lowest molecular weight of the seven factors examined, and consequently may cause the least steric hindrance. Consistent with this, SunTag with a short linker between the tags yields poorer activation as a result of steric hindrance in our previous work [16]. A linker of 5 amino acids, 22 amino acids, and 43 amino acids were compared in the SunTag system with TET1. The 22-amino acid linker gives the best result and the 5-amino acid linker is the worst [16]. Thus, different linker lengths could change the activity of the factors. Judging from the fact that the molecular weight of TET1 is greater than all the factor X including VP64, which is the smallest among the factor X, the 22-amino acid linker would also give the best result in the SunTag system with TET1 and VP64.

Activated expression levels vary among genes. The expression level of each gene could be affected by several factors such as, structure of the promoter and efficiency of gRNAs. The expression levels observed in the SunTag system with TET1 and VP64 showed mild correlation with CpG observed/expected (O/E) ratio around sgRNAs (R = 0.57). This suggests that CpG-rich promoters tend to be more activated.

Our SunTag system yielded better activation than the direct fusion system (dCas9–TET1 and dCas9–VP64) (Figure 4) using the same single sgRNA. The activation range of the direct fusion system is low when using a single sgRNA; consequently, tiling with multiple sgRNAs is usually necessary to achieve robust transcriptional activation [18]. Thus, one of the merits of our system is that only a single sgRNA is required for robust activation. This makes the system simple and versatile, and thus very useful for genome-wide screening or therapeutic applications. Another merit of our system is the shorter length of the constructs, due to the relatively small size of the SunTag. Constructs for the fusion system must be at least the sum of the lengths of the dCas9 and TET1 genes. Unfortunately, in this regard, dCas9 (4.2 kb) and catalytic domain of TET1 (2.2 kb) are long DNAs, and this limits their use in gene therapy, which usually uses viral vectors with limited packaging capacity (4.7 kb for AAV). Even using shorter dCas9 from *Staphylococcus aureus* (dSaCas9, 3.2 kb), dSaCas9-TET1 (totally 5.4 kb) exceeds the size limit of the AAV vector (4.7 kb). On the other hand, the length of a construct for the SunTag system is the sum of the lengths of dCas9 and SunTag (0.5 kb), the latter of which is quite short.

Overexpression of endogenous genes is becoming increasingly important for therapeutic applications. For example, overexpression of a functional endogenous copy has the potential to rescue human diseases caused by haploinsufficiency [19]. Alternatively, overexpression of a protein similar to one encoded by a mutant gene could treat human diseases caused by recessive mutations [20]. For example, congenital muscular dystrophy type 1A (MDC1A) is an autosomal recessive disorder caused by mutations in *LAMA2* that cause production of nonfunctional laminin-α2. Viral overexpression of *Lama1*, which encodes a similar protein, in a mouse model of MDC1A improved disease symptoms and slowed progression [20]. Thus, the development of a system that yields robust activation and yet is small enough to be cloned into a virus vector, is an important priority for therapeutic applications. Therefore, we anticipate that our modified SunTag system will facilitate advances in gene therapy.

## 4. Materials and Methods

### 4.1. Construction of sgRNAs

For each target gene, a unique 20 bp sequence was selected around the transcription start site. Selected gRNAs were cloned under control of the human U6 promoter (gRNA_Cloning Vector BbsI, Addgene 128433). The target sequences are described in the Appendix A.

### 4.2. Cell Culture

A549 (RIKEN BRC, Tsukuba, Japan) cells were cultured at 37 °C under 5% CO_2_ in minimum essential medium (MEM) (M4655-500ML, Sigma-Aldrich, St. Louis, MO, USA) supplemented with 10% fetal bovine serum (FBS) and non-essential amino acids. A549 cells were transfected with Lipofectamine 2000 (Invitrogen, Carlsbad, CA, USA). Blasticidin-S (Invitrogen, Carlsbad, CA, USA) was added to the culture medium at a final concentration of 2 μg/mL 48 h after transfection. After 3 days of selection, cells were harvested and subjected to analysis. The molar ratio of the dCas9–SunTag vector, scFv–TET1 vector and scFv–X vector to sgRNA vector in the transfection was 1:1:1:1.5, respectively. In case of the direct fusion system, the molar ratio of the dCas9–TET1 vector and dCas9–VP64 vector to sgRNA vector in the transfection was 1:1:1.5, respectively. All constructs except for the sgRNA vector were expressed under the control of the CAG promoter.

### 4.3. Quantitative RT-PCR Analysis

Total RNA was isolated from cells using the AllPrep DNA/RNA micro kit (Qiagen, Hilden, Germany). Gene expression was measured with a LightCycler 98 (Roche, Roche, Basel, Switzerland) using TB Green Premix Ex Taq II (Takara, Kusatsu, Japan). Expression levels were normalized against the corresponding level of actin mRNA and were presented as fold changes relative to GFP-transfected experimental controls. Primer sequences are described in the Appendix A.

## Figures and Tables

**Figure 1 ijms-21-01574-f001:**
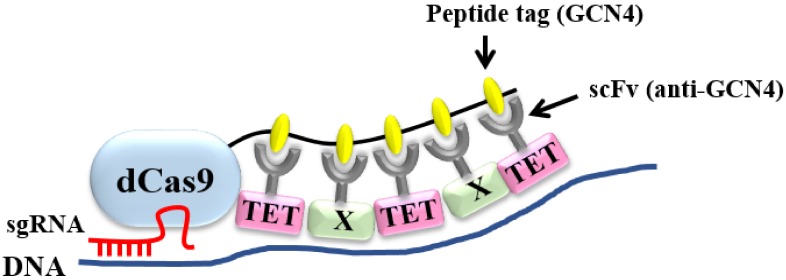
Modified dCas9–SunTag format for simultaneous recruitment of TET1 and another factor (Factor X). In the modified dCas9–SunTag format, dCas9 is fused to a tandem array of multiple copies of GCN4 peptide tag separated by a 22-amino acid linker. This format can recruit multiple copies of fusion proteins of anti-GCN4 peptide antibody (scFv) with TET1 (scFv–TET1) and Factor X (scFv–X), which should synergistically activate the target gene.

**Figure 2 ijms-21-01574-f002:**
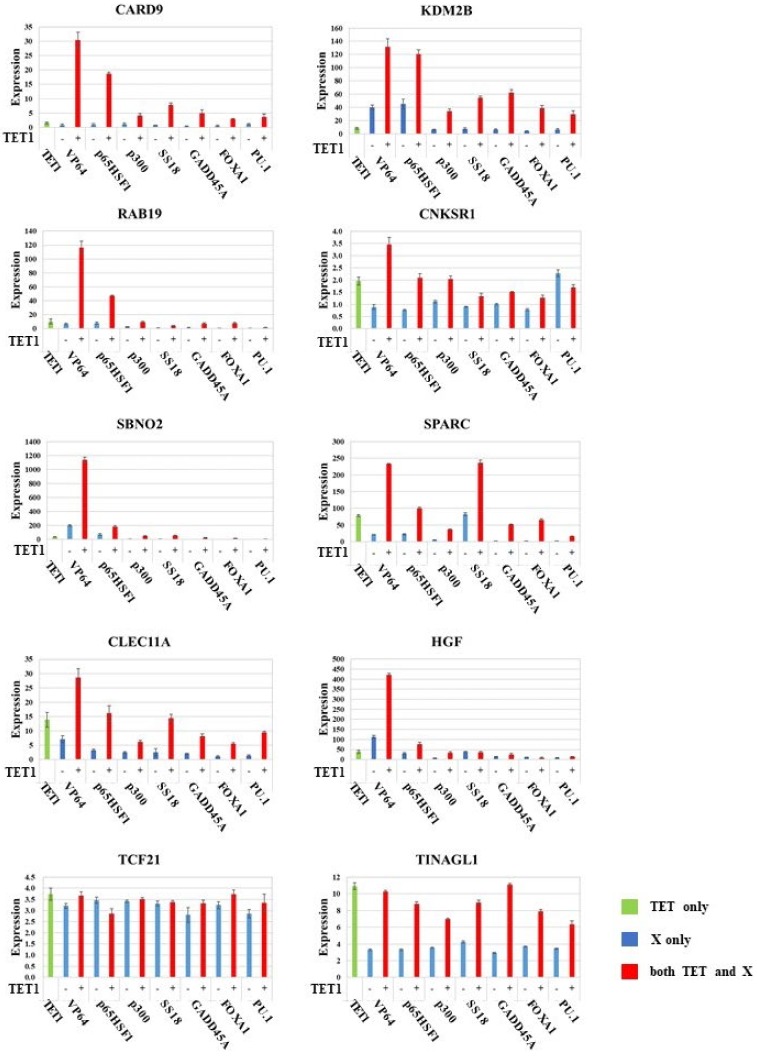
Expression levels in A549 cells transfected with dCas9–SunTag and scFv–TET1 (green bar); dCas9–SunTag and scFv–X (blue bar); and dCas9–SunTag, scFv–TET1, and scFv–X (red bar). Expression levels, determined by RT-PCR, were normalized against the corresponding level of actin mRNA and are presented as fold changes relative to GFP (green fluorescent protein)-transfected experimental controls. Results obtained using sgRNAs targeting *CARD9*, *KDM2B*, *RAB19*, *CNKSR1*, *SBNO2*, *SPARC*, *CLEC11A*, *HGF*, *TCF21*, and *TINAGL1* are shown.

**Figure 3 ijms-21-01574-f003:**
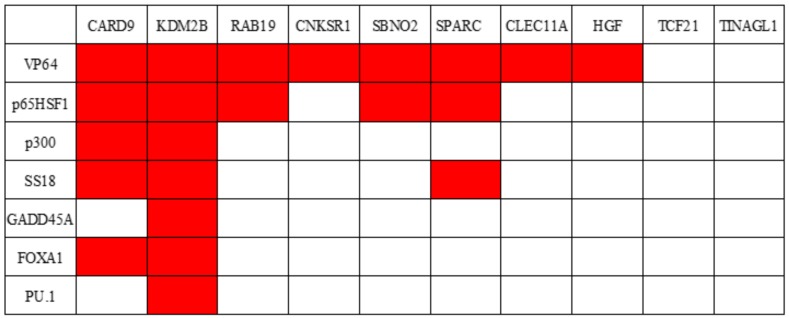
Synergy of TET1 and other factors in the modified SunTag format. Red cells indicate significant synergy between TET1 and another factor (*p* < 0.05). Each row indicates a factor (VP64, p65HSF1, p300, SS18, GADD45A, FOXA1, or PU.1), and each column indicates a gene (*CARD9*, *KDM2B*, *RAB19*, *CNKSR1*, *SBNO2*, *SPARC*, *CLEC11A*, *HGF*, *TCF21*, or *TINAGL1*).

**Figure 4 ijms-21-01574-f004:**
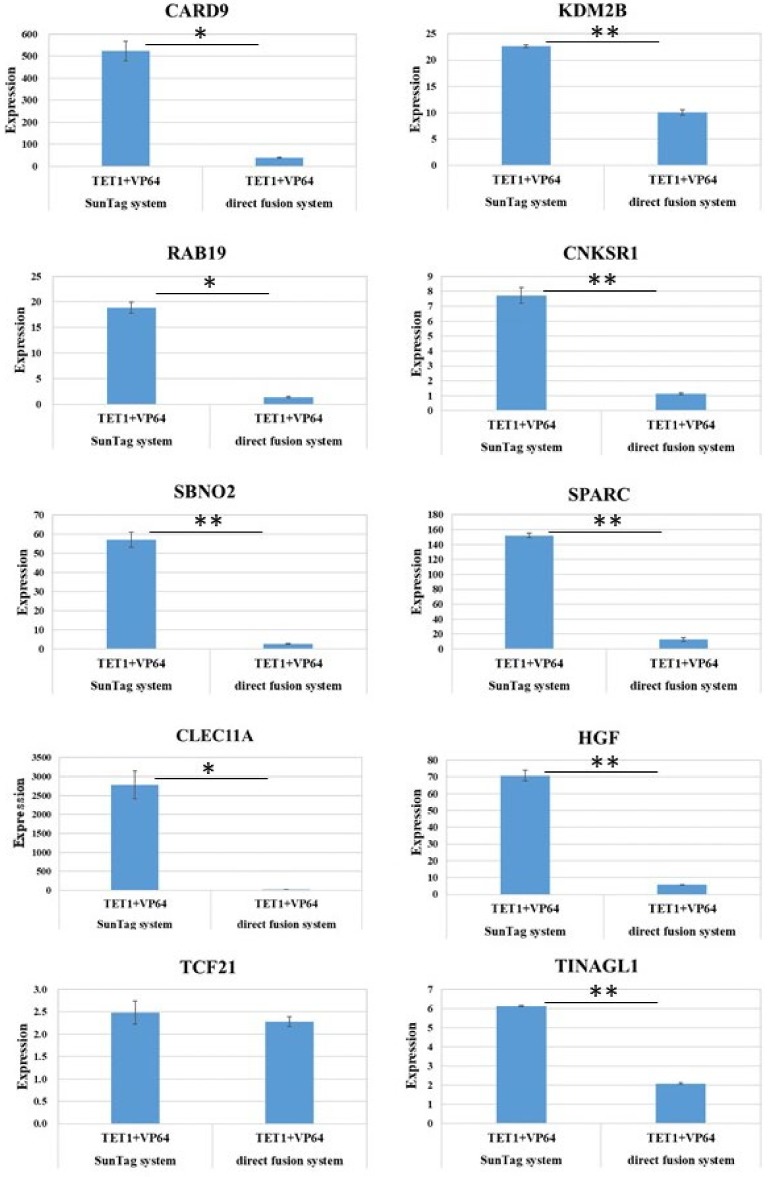
Comparison of synergy in modified SunTag and dCas9 direct fusion. Expression levels are shown for cells transfected with dCas9–SunTag, scFv–TET1, and scFv–VP64 (left); or dCas9–TET1 and dCas9–VP64 (right). Expression levels, as determined by RT-PCR, are normalized against the level of actin and presented as fold changes relative to GFP-transfected experimental controls. Results using sgRNAs for *CARD9*, *KDM2B*, *RAB19*, *CNKSR1*, *SBNO2*, *SPARC*, *CLEC11A*, *HGF*, *TCF21*, and *TINAGL1* are shown. * *p* < 0.05, ** *p* < 0.01.

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
