# Peer review of "Synergistic Upregulation of Target Genes by TET1 and VP64 in the dCas9–SunTag Platform"

_ijms, 2020, doi:10.3390/ijms21051574_

Round 1
Reviewer 1 Report
The manuscript by Sumiyo Morita and colleagues describes an optimized version of the dCas9-SunTag system that can be used to activate gene expression using a single sgRNA. The data is presented in a concise and clear manner, but there are a few points that need revisiting before this manuscript should be accepted for publication.
In paragraph 2.1 they do describe the modified version but they do not elaborate on how the original system was modified. From the introduction I gather that they optimized the linker length but they do not mention any specifics nor do they show the experiments they performed to prove that this optimized version is working better than the original. If this has been shown in a previous publication then this part should not be included in the results section but should rather be referred to in the introduction. Figure 2: for the ease of the reader it would be helpful to label the red, blue and green bars. E.g: green = TET only, blue =X only, red = both Discussion: The sentence in line 126/127 refers to the effect of different linker lengths on activity of the SunTag: similar to what was mentioned in point 1, it would be nice to see some experiments investigating how different linker lengths will change the activity of the different Factors. This might for example be one explanation why VP64 was performing the best as that particular linker length as it might be suited best for that combination. Sentence in line 139/140 (Another merit of our system ….) is similar to the sentence in 144/145 and hence I think one could be removed. There is reference throughout the manuscript that this system would be suitable for packaging into viral vectors and hence can be used for gene therapy. It would be helpful to give the reader some sense of size. For example: how much smaller is this SunTag system compared to the TET1-VP64 fusion? What are the size limits for AAV packaging? Methods 4.1: information is missing about which gRNA vector was used for cloning the gRNAs Methods 4.2: There is no mention of the GFP transfection as experimental controls Methods 4.3: the data is presented as fold changes relative to GFP-transfected experimental controls. It is unclear to me whether this refers to co-transfection of GFP or a separate well that was transfected with GFP. If it was the letter then transfection efficiencies between this GFP sample and the experimental sample might be different and hence the data should not be shown as fold changes relative to GFP. This needs clarification.Author Response
Comment1: In paragraph 2.1 they do describe the modified version but they do not elaborate on how the original system was modified. From the introduction I gather that they optimized the linker length but they do not mention any specifics nor do they show the experiments they performed to prove that this optimized version is working better than the original. If this has been shown in a previous publication then this part should not be included in the results section but should rather be referred to in the introduction.
Response1: We sincerely apologize for any confusion. In our previous report (Ref.16), we optimized the linker length between the tags to maximize transcriptional activation by the effector TET1. We described this system as modified SunTag. We think independent paragraph 2.1 could cause confusion. Therefore, we removed the title of paragraph 2.1 and combined with paragraph 2.2.
Comment2: Figure 2: for the ease of the reader it would be helpful to label the red, blue and green bars. E.g: green = TET only, blue =X only, red = both
Response2: We greatly appreciate the reviewer’s insightful comments about our study, which have helped us to significantly improve our paper. We added the label according to the reviewer’s suggestion (Fig. 2).
Comment3: Discussion: The sentence in line 126/127 refers to the effect of different linker lengths on activity of the SunTag: similar to what was mentioned in point 1, it would be nice to see some experiments investigating how different linker lengths will change the activity of the different Factors. This might for example be one explanation why VP64 was performing the best as that particular linker length as it might be suited best for that combination.
Response3: We appreciate the reviewer’s helpful advice. In our previous work (Ref.16), a linker of 5 amino acid, 22 amino acid, and 43 amino acid were compared in SunTag system with TET1. The 22- amino acid linker gives the best result and 5- amino acid linker is the worst. Thus, different linker lengths could change the activity of the factors as you indicated. Judging from the fact that molecular weight of TET1 is greater than all the factor X (VP64, p65HSF1, p300, SS18, GADD45A, FOXA1, and PU.1) including VP64 which is the smallest among the factor X, 22-amino acid linker would also give the best result in the SunTag system with TET1 and VP64. We have added following descriptions in Discussion: “A linker of 5 amino acid, 22 amino acid, and 43 amino acid were compared in SunTag system with TET1. The 22- amino acid linker gives the best result and 5- amino acid linker is the worst (Ref.16). Thus, different linker lengths could change the activity of the factors. Judging from the fact that moleculer weight of TET1 is greater than all the factor X including VP64 which is the smallest among the factor X, 22-amino acid linker would also give the best result in the SunTag system with TET1 and VP64.” (Line 131-136)
Comment4: Sentence in line 139/140 (Another merit of our system ….) is similar to the sentence in 144/145 and hence I think one could be removed. There is reference throughout the manuscript that this system would be suitable for packaging into viral vectors and hence can be used for gene therapy. It would be helpful to give the reader some sense of size. For example: how much smaller is this SunTag system compared to the TET1-VP64 fusion? What are the size limits for AAV packaging?
Response4: We appreciate the reviewer’s helpful advice. We have modified the sentences in Discussion as follows: “Another merit of our system is the shorter length of the constructs, due to the relatively small size of the SunTag. Constructs for the fusion system must be at least the sum of the lengths of the dCas9 and TET1 genes. Unfortunately, in this regard, dCas9 (4.2 kb) and TET1 (2.2 kb) are long genes, and this limits their use in gene therapy, which usually uses viral vectors with limited packaging capacity (4.7 kb for AAV). Even using shorter dCas9 from Staphylococcus aureus (3.4 kb), dCas9 (3.4 kb) -TET1 (2.2 kb) (totally 5.6 kb) exceeds the size limit of AAV vector (4.7kb). On the other hand, the length of a construct for the SunTag system is the sum of the lengths of dCas9 and SunTag (0.5 kb), the latter of which is quite short.” (Line 154-162)
Comment5: Methods 4.1: information is missing about which gRNA vector was used for cloning the gRNAs
Response5: We appreciate the reviewer’s helpful advice. We added the name of the sgRNA vector in Materials and Methods in accordance with the reviewer’s suggestion. (Line 177-178)
Comment6: Methods 4.2: There is no mention of the GFP transfection as experimental controls
Response6: We appreciate the reviewer’s helpful advice. We have added following descriptions in Results: “Expression levels were normalized against the corresponding level of actin mRNA and are presented as fold changes relative to GFP-transfected experimental controls.” (Line 91-92)
Comment7: Methods 4.3: the data is presented as fold changes relative to GFP-transfected experimental controls. It is unclear to me whether this refers to co-transfection of GFP or a separate well that was transfected with GFP. If it was the letter then transfection efficiencies between this GFP sample and the experimental sample might be different and hence the data should not be shown as fold changes relative to GFP. This needs clarification.
Response7: We appreciate the reviewer’s helpful advice. GFP was transfected as separate well. We did not normalize expression levels with GFP. Expression levels were normalized against the corresponding level of actin mRNA and are presented as fold changes relative to separately GFP-transfected experimental controls. Presentation as fold changes relative to GFP-transfected experimental controls is usually used as in Reference 6. We have added following descriptions in Results: “Expression levels were normalized against the corresponding level of actin mRNA and are presented as fold changes relative to GFP-transfected experimental controls.” (Line 91-92)
Reviewer 2 Report
Authors of the manuscript performed a functional screen of a set of transcription activator/modulator domains to identify the best candidate to achieve synergistic activation of target genes expression in combination with characterized in the previous study the dCas9-SunTag platform containing the effector TET1. Presented data, showing a robust, synergistic activation of the expression of the majority of the examined genes with the combination of TET1 and VP64, are very interesting and potentially impactful. There are however some minor concerns that would need to be addressed to raise the quality of the manuscript sufficiently:
It is not clear why this particular set of genes was chosen. Are these genes normally epigenetically silent in the cell line chosen? Are they potential therapeutic anti-cancer targets (the cell line is lung derived carcinoma)? Were they chosen because of the presence of particular regulatory motifs in their promoters? Can the structures of the promoter regions explain wide range of responses shown in Fig 2? This should be explained in Result and/or Discussion sections.
What are the steady-state levels of expression of these genes in A549 cells? Is there a correlation between the level of expression and the level of dCas9-SunTag platform induced activation? (Discussion) Can you provide in Supplementary data showing relative expression levels (target/reference ratios) of the examined genes?
Why fold changes for the same genes in Fig 2 and 4 are not matching? For example, for CARD9, the fold change in Fig 2 in the presence of TET1 and VP64 is ~30, but in Fig 4 for the same condition, it is over 500?
Can you explain what “and the cells that exhibited a higher expression level were selected.” in Line 94 means?
As mentioned, different activation domains had different sizes. Was the total amount of DNA in all transfections equilibrated?
Author Response
Comment1: It is not clear why this particular set of genes was chosen. Are these genes normally epigenetically silent in the cell line chosen? Are they potential therapeutic anti-cancer targets (the cell line is lung derived carcinoma)? Were they chosen because of the presence of particular regulatory motifs in their promoters? Can the structures of the promoter regions explain wide range of responses shown in Fig 2? This should be explained in Result and/or Discussion sections.
Response1: We greatly appreciate the reviewer’s insightful comments about our study, which have helped us to significantly improve our paper. We chose these genes because these are hypermethylated in the cell line (A549: lung adenocarcinoma cell line) as you indicated. Therefore, they could be potential therapeutic anti-cancer targets. We did not choose by any regulatory motifs in the promoter. The expression level of each gene could be affected by several factors such as, structure of the promoter and efficiency of gRNAs. The expression levels observed in SunTag system with TET1 and VP64 showed mild correlation with CpG observed/expected (O/E) ratio around sgRNAs (R = 0.57). This suggests that CpG-rich promoters tend to be more activated. We have added following descriptions in Results: “All these genes are hypermethylated in A549 lung adenocarcinoma cell line.” (Line 90-91) We have also added following descriptions in Discussion: “Activated expression levels vary among genes. The expression level of each gene could be affected by several factors such as, structure of the promoter and efficiency of gRNAs. The expression levels observed in SunTag system with TET1 and VP64 showed mild correlation with CpG observed/expected (O/E) ratio around sgRNAs (R = 0.57). This suggests that CpG-rich promoters tend to be more activated.” (Line 144-148)
Comment2: What are the steady-state levels of expression of these genes in A549 cells? Is there a correlation between the level of expression and the level of dCas9-SunTag platform induced activation? (Discussion) Can you provide in Supplementary data showing relative expression levels (target/reference ratios) of the examined genes?
Response2: Basically, all these genes are hypermethylated and the expression levels are low in A549 cells. Relative expression levels (expression level normalized by actin) are provided as Supplementary Figure 1. There seems to be no correlation between steady-state levels and the induced levels.
Comment3: Why fold changes for the same genes in Fig 2 and 4 are not matching? For example, for CARD9, the fold change in Fig 2 in the presence of TET1 and VP64 is ~30, but in Fig 4 for the same condition, it is over 500?
Response3: Basically, all these genes are hypermethylated and the expression levels are low in A549 cells. Therefore, baseline of each gene (expression level in GFP-transfected A549 cells) is very low and variation is large. Thus, value normalized by baseline vary.
Comment4: Can you explain what “and the cells that exhibited a higher expression level were selected.” in Line 94 means?
Response4: I apologize for the difficult explanation. Synergy should be examined by the comparison between C and higher one of A and B. If A is higher than B. A is selected and compared with C. If B is higher than A. B is selected and compared with C.
A: The expression level in cells transfected with dCas9–SunTag and scFv–X (Figure 2, blue bar).
B:The expression level in cells transfected with dCas9–SunTag and scFv–TET1 (Figure 2, green bar)
C:The expression level in cells transfected with dCas9–SunTag, scFv–TET1, and scFv–X (Figure 2, red bar)
Comment5: As mentioned, different activation domains had different sizes. Was the total amount of DNA in all transfections equilibrated?
Response5: Yes, we transfected equimolar amount of each DNA. This was described in Materials and Methods as follows: “The molar ratio of the dCas9–SunTag vector, scFv–TET1 vector and scFv–X vector to sgRNA vector in the transfection was 1:1:1:1.5, respectively.” (Line 185-186)
Reviewer 3 Report
The manuscript entitled “Synergistic upregulation of target genes by TET1 and VP64 in the dCas9–SunTag platform” reported synergistic activation of endogenous gene expression via simultaneous tethering of VP64 and TET1 to dCas9–SunTag. The modified system was applied to different loci and desirable gene activation was observed. I have the following concerns:
Figure 4 showed that the modified SunTag system could achieve significantly higher level of gene activation than direct fusion system. Since the same sgRNA was used for both dCas9–TET1 and dCas9–VP64, it’s more likely that only one could bind to the targeting site and function. Do you think it will be fairer to use different sgRNAs in close proximity for both complexes when comparing two systems? Also, I didn’t see details on how to perform the direct fusion assay in the method part. As a method paper, I believe these information will be necessary for readers. Figure 3 merely describes the statistical significance of results shown in Figure 2. To keep this communication paper concise, this part could be removed by indicating the statistical significance in Figure 2. I think it would be easier to read if the authors add some background on how VP64 and TET1 activate gene expression in the Introduction part. Too much background and too few results in the Abstract.Author Response
Comment1: Figure 4 showed that the modified SunTag system could achieve significantly higher level of gene activation than direct fusion system. Since the same sgRNA was used for both dCas9–TET1 and dCas9–VP64, it’s more likely that only one could bind to the targeting site and function. Do you think it will be fairer to use different sgRNAs in close proximity for both complexes when comparing two systems?
Response1: We greatly appreciate the reviewer’s insightful comments about our study, which have helped us to significantly improve our paper. It is difficult to use the format with multiple sgRNAs for genome-wide gain-of-function screening and gene therapy. The format with a single sgRNA is simple and versatile, and thus very useful for genome-wide screening or therapeutic applications. Therefore, we compared the systems with a single sgRNA. In addition, once dCas9–TET1 binds to the target and demethylate it, hypomethylated state can be maintained after dCas9–TET1 left the target in dynamic equilibrium. Therefore, dCas9–VP64 can bind to the hypomethylated target and activate gene expression with a single sgRNA.
Comment2: Also, I didn’t see details on how to perform the direct fusion assay in the method part. As a method paper, I believe these information will be necessary for readers.
Response2: We thank the reviewer for this comment. All the experimentally procedure for the direct fusion is same as that of SunTag system. To clarify this, we have added following descriptions in Materials and Methods: “In case of direct fusion system, the molar ratio of the dCas9–TET1 vector and dCas9–VP64 vector to sgRNA vector in the transfection was 1:1:1.5, respectively.” (Line 187-188)
Comment3: Figure 3 merely describes the statistical significance of results shown in Figure 2. To keep this communication paper concise, this part could be removed by indicating the statistical significance in Figure 2.
Response3: We appreciate your helpful comment. Synergy between TET1 and X was judged as follows. The expression level in cells transfected with dCas9–SunTag and scFv–X (Figure 2, blue bar) was compared with the level in cells transfected with dCas9–SunTag and scFv–TET1 (Figure 2, green bar), and the cells that exhibited a higher expression level were selected. These expression levels were compared with those of cells transfected with dCas9–SunTag, scFv–TET1, and scFv–X (Figure 2, red bar). The results are summarized in Figure 3.
If these comparisons are described in Figure 2, this figure becomes very busy. Therefore, we purposely summarized the comparisons in Figure 3.
Comment4: I think it would be easier to read if the authors add some background on how VP64 and TET1 activate gene expression in the Introduction part. Too much background and too few results in the Abstract.
Response4: We added following description in Abstract according to the reviewer’s suggestion: “Here we describe the screening of factors that exhibit the best synergistically activation of gene expression with TET1 in the dCas9–SunTag format. All seven factors examined show some synergy with TET1. Among them, VP64 gives the best results. Thus, simultaneous tethering of VP64 and TET1 to a target gene using an optimized dCas9–SunTag format synergistically activates gene expression using a single sgRNA.” (Line 16-20)
Reviewer 4 Report
In their manuscript “Synergistic upregulation of target genes by TET1 and VP64 in the dCas9–SunTag platform" Sumiyo Morita and colleagues describe a dCas9-SunTag platform to synergistic upregulate the expression of target genes. They Modified dCas9–SunTag format for simultaneous recruitment of TET1 and different factors, and compared synergistic effects between modified SunTag and dCas9 direct fusion to suggest that the dCas9–SunTag system can be used to upregulate target genes.
Concern 1
Result
2.2. Selection of another factor that activates gene expression synergistically with TET1 in the modified dCas9–SunTag format
Ten genes (CARD9, KDM2B, RAB19, CNKSR1, SBNO2, SPARC, CLEC11A, HGF, TCF21, and TINAGL1) were examined for each factor.
Why did the authors select these genes as targets? Can the selected gene represent the expression pattern of most of genes, and enough to support the conclusion that “relative to other factors, transcription factors yielded a stronger synergistic effect”?
Concern 2
2.3. Comparison of synergistic effects between modified SunTag and dCas9 direct fusion
Ten genes (CARD9, KDM2B, RAB19, CNKSR1, SBNO2, SPARC, CLEC11A, HGF, TCF21, and TINAGL1) were examined at the beginning, but only three (CARD9, KDM2B, RAB19) were selected to compare the synergistic effects between modified SunTag and dCas9 direct fusion, this point should be clarified. Whether the result from three genes can have the conclusion that the SunTag system yielded significantly better activation compare to dCas9 direct fusion.
Concern 3
Discussion
The authors conclude that “Another merit of our system is the shorter length of the constructs, due to the relatively small size of the SunTag. Constructs for the fusion system must be at least the sum of the lengths of the dCas9 and TET1 genes.” and “On the other hand, the length of a construct for the SunTag system is the sum of the lengths of dCas9 and SunTag, the latter of which is quite short.”
I understood that the study was following the previous work which has been published, but a supplementary file, including the sequence of the SunTag system, is necessary to support this point.
Author Response
Concern 1
Result
2.2. Selection of another factor that activates gene expression synergistically with TET1 in the modified dCas9–SunTag format
Ten genes (CARD9, KDM2B, RAB19, CNKSR1, SBNO2, SPARC, CLEC11A, HGF, TCF21, and TINAGL1) were examined for each factor.
Why did the authors select these genes as targets? Can the selected gene represent the expression pattern of most of genes, and enough to support the conclusion that “relative to other factors, transcription factors yielded a stronger synergistic effect”?
Response1: We greatly appreciate the reviewer’s insightful comments about our study, which have helped us to significantly improve our paper. TET1 performs the first enzymatic steps during DNA demethylation. Therefore, we chose these genes because these are hypermethylated in A549 cell line (lung adenocarcinoma cell line). We have added following descriptions in Results: “All these genes are hypermethylated in A549 lung adenocarcinoma cell line.” (Line 90-91)
Concern 2
2.3. Comparison of synergistic effects between modified SunTag and dCas9 direct fusion
Ten genes (CARD9, KDM2B, RAB19, CNKSR1, SBNO2, SPARC, CLEC11A, HGF, TCF21, and TINAGL1) were examined at the beginning, but only three (CARD9, KDM2B, RAB19) were selected to compare the synergistic effects between modified SunTag and dCas9 direct fusion, this point should be clarified. Whether the result from three genes can have the conclusion that the SunTag system yielded significantly better activation compare to dCas9 direct fusion.
Response2: We thank the reviewer for this comment. We added additional data for remaining seven genes in Fiugre 4.
Concern 3
Discussion
The authors conclude that “Another merit of our system is the shorter length of the constructs, due to the relatively small size of the SunTag. Constructs for the fusion system must be at least the sum of the lengths of the dCas9 and TET1 genes.” and “On the other hand, the length of a construct for the SunTag system is the sum of the lengths of dCas9 and SunTag, the latter of which is quite short.”
I understood that the study was following the previous work which has been published, but a supplementary file, including the sequence of the SunTag system, is necessary to support this point.
Response3: To make the sentences easy to understand, we have modified the sentences in Discussion as follows: “Another merit of our system is the shorter length of the constructs, due to the relatively small size of the SunTag. Constructs for the fusion system must be at least the sum of the lengths of the dCas9 and TET1 genes. Unfortunately, in this regard, dCas9 (4.2 kb) and TET1 (2.2 kb) are long genes, and this limits their use in gene therapy, which usually uses viral vectors with limited packaging capacity (4.7 kb for AAV). Even using shorter dCas9 from Staphylococcus aureus (3.4 kb), dCas9 (3.4 kb) -TET1 (2.2 kb) (totally 5.6 kb) exceeds the size limit of AAV vector (4.7kb). On the other hand, the length of a construct for the SunTag system is the sum of the lengths of dCas9 and SunTag (0.5 kb), the latter of which is quite short.” (Line 154-162)
Round 2
Reviewer 4 Report
The authors have addressed all my comments.